# Daphnetin, a Coumarin with Anticancer Potential against Human Melanoma: In Vitro Study of Its Effective Combination with Selected Cytostatic Drugs

**DOI:** 10.3390/cells12121593

**Published:** 2023-06-09

**Authors:** Paula Wróblewska-Łuczka, Agnieszka Góralczyk, Jarogniew J. Łuszczki

**Affiliations:** Department of Occupational Medicine, Medical University of Lublin, ul. Jaczewskiego 8b, 20-090 Lublin, Poland; paula.wroblewska-luczka@umlub.pl (P.W.-Ł.); agnieszka.goralczyk@umlub.pl (A.G.)

**Keywords:** melanoma, coumarins, daphnetin, chemotherapeutics, isobolographic analysis

## Abstract

(1) The treatment of metastatic or drug-resistant melanoma is still a significant therapeutic problem. The aim of this study was to evaluate the anticancer potential of daphnetin (7,8-dihydroxycoumarin) and its combinations with five different cytostatic drugs (mitoxantrone, docetaxel, vemurafenib, epirubicin and cisplatin). (2) The viability, proliferation and cytotoxicity of daphnetin against four human malignant melanoma cell lines were evaluated. The interactions were assessed using isobolographic analysis for the combinations of daphnetin with each of the five cytostatic drugs. (3) Daphnetin showed anticancer activity against malignant melanoma, with IC_50_ values ranging from 40.48 ± 10.90 µM to 183.97 ± 18.82 µM, depending on the cell line. The combination of daphnetin with either vemurafenib or epirubicin showed an antagonistic interaction. Moreover, additive interactions were observed for the combinations of daphnetin with cisplatin and docetaxel. The most desirable synergistic interactions for human melanoma metastatic cell lines were observed for the combination of daphnetin with mitoxantrone. (4) The obtained results suggest that daphnetin should not be combined with vemurafenib or epirubicin in the treatment of malignant melanoma due to the abolition of their anticancer effects. The combination of daphnetin with mitoxantrone is beneficial in the treatment of metastatic melanoma due to their synergistic interaction.

## 1. Introduction

Melanoma can appear anywhere on the body where there are pigment cells—melanocytes–including in mucous membranes, eyeballs and hair follicles. However, it most often appears on the skin. Melanoma is a cancer that has a high survival rate but only when it is diagnosed early. Compared with other skin cancers, melanoma patients are more likely to have metastases. As the stage of the cancer increases, the survival rate of patients decreases [1,2]. For several decades, the incidence of melanoma has been gradually increasing. This phenomenon is observed all over the world. Ethnic origin and geographical location, but also age and gender, have a significant impact on the melanoma incidence. The annual increase in the number of cases is notable especially in light-skinned populations. This is probably related to the reduced amount of melanin, and thus reduced photoprotection, in the skin of Caucasian populations [3,4].

Early diagnosis of melanoma allows for effective treatment (mainly by surgical methods) in its early stages [4]. When the disease is already at an advanced stage and metastasis to distant organs is evident, more invasive methods of treatment are used. These include chemotherapy, radiotherapy, immunotherapy, photodynamic therapy (PDT) and targeted therapy. However, the drugs used often have a number of side effects, and after prolonged use, resistance to the drug therapy appears [2,4,5].

Nowadays, it has become popular to search for new therapeutic substances of natural origin. Active substances of plant origin may be the foundation of less invasive treatments used in the future, especially due to the lack of or mild side effects. Each of the natural compounds is characterized by a different chemical structure and various biological activities. An interesting group of compounds are coumarins; it is estimated that, so far, about 1300 substances of this group have been isolated from plants, fungi or bacteria [6,7].

One of the simple coumarins is daphnetin (7,8-dihydroxycoumarin), which was first isolated from plants of the genus *Daphne*, from which the name of a compound is derived. *Daphne* plants are flowering shrubs (comprising several dozen species) found in Europe, Asia and North Africa [8]. Daphnetin is a natural substance derived from these plants; however, unfortunately, this is a fact that limits the acquisition and use of the compound on a larger scale. Chemists have developed methods to synthesize daphnetin to facilitate and increase its production [7]. Daphnetin is a highly soluble compound (especially in organic solvents), and in the body it is metabolized by cytochrome P450 3A4 [9].

Daphnetin possesses a number of biological activities, including anti-inflammatory, antioxidant, antibacterial, antidiabetic, immunosuppressive and anticancer activities [7,10]. Neuroprotective effects have also been proven in many in vitro and animal experiments. Neuroprotection results from the strong antioxidant activity of daphnetin, especially where oxidative stress, reactive oxygen species and mediators of the inflammatory process have resulted in the development of neurodegenerative diseases. It has been proven that daphnetin increases the amount of antioxidants (GSH and SOD) and reduces the levels of caspase-3 and pro-inflammatory cytokines. The JNK-MAPK, JAK-STAT and TLR-4/NF-κB signaling pathways inhibited by daphnetin cause an increase in the anti-apoptotic Bcl-2 proteins with a simultaneous decrease in Bax proteins, which also contribute to the neuroprotective effect [11]. Daphnetin also has potential in the treatment of diabetes due to its properties that stimulate the release of insulin. Daphnetin improved the viability of rat insulinoma (INS-1) cells that were later treated with streptozotocin. Pre-treatment of cells with coumarin improved the activity of antioxidant enzymes and reduced the level of lipid peroxidation [12]. In silico molecular modeling has demonstrated the potential of daphnetin in the treatment of COVID-19 infection. Daphnetin, due to its structure, shows good potential binding to the NSP10/NSP16 methyltransferase stimulating-factor complex of the SARS-CoV-2 virus, which protects the mRNA transcripts of the virus against the immune system [13].

Daphnetin has a strong antibacterial effect, which has already been used in traditional Chinese medicine. It is important that bacteria do not develop resistance to the compound after its application. Experiments have shown that daphnetin is effective in neutralizing the dangerous bacterium *Helicobacter pylori*. The mechanism of action is based on the release of specific proteins, the binding of ions and the prevention of colonization by *H. pylori*. This is of great importance for preventing bacterial infection but also, due to the colonization of the stomach, preventing the development of peptic ulcers or cancer of this organ [14]. The use of daphnetin may also indirectly prevent the development of cervical cancer due to the ability of this compound to bind to the protein of the inflammatory process, HMGB1, the release of which is observed in connection with infection caused by the oncogenic virus HPV-18 [15].

## 2. Materials and Methods

### 2.1. Malignant Melanoma Cell Culture

In this experiment, we used two metastatic (FM55M2 and SK-MEL28) and two primary (FM55P and A375) melanoma cell lines. From the European Collection of Authenticated Cell Cultures (ECACC, Public Health England, Porton Down, UK), we purchased FM55P and FM55M2 cell lines, and from the American Type Culture Collection (ATCC, Manassas, VA, USA), we purchased A375 and SK-MEL28 cell lines. Of note, the A375 cell line is the original and not a CRISPR-modified version of the cell line, and all the tested cell lines in this study have the BRAF V600E mutation. Experimental cell culture conditions: 37 °C in a humidified atmosphere of 95% air and 5% CO_2_. Culture media used: for A375—DMEM high glucose; for SK-MEL28—EMEM; and for FM55P and FM55M2—RPMI1640. Previous publications have described in more detail the growth conditions of these cell line cultures [16,17].

### 2.2. Tested Drugs

Daphnetin (DAP) and the chemotherapeutics—mitoxantrone (MTX), docetaxel (DOCX) and vemurafenib (VEM)—were dissolved in DMSO. Other test compounds such as epirubicin (EPR) were dissolved in sterile hot water, and cisplatin (CDDP) was dissolved in PBS buffer (all the drugs were from Sigma-Aldrich, St. Louis, MO, USA). Compounds were dissolved to the tested concentrations in growth medium before adding compounds to the microtiter (96-well) plate. DMSO was maintained at the safe concentration of 0.1% and had no effect on cell growth.

### 2.3. Cell Viability Assessment—MTT Test

The effect on cell viability of DAP and the five chemotherapeutics (MTX, DOCX, VEM, EPR and CDDP) was assessed using the MTT assay. All tested cell lines—the HaCaT cell line model of human keratinocyte cells [18] (density: 1 × 10^4^ cells/mL), HEMa-LP melanocytes (density: 5 × 10^3^ cells/mL) and four human malignant melanoma lines (density: 2–3 × 10^4^ cells/mL, depending on the cell line—were plated on microtiter plates (NEST Biotechnology, Wuxi, China). After a day (24 h) of incubation, the medium was replaced with fresh medium to which was added increasing concentrations of the tested substances (DAP, MTX, DOCX, VEM, EPR and CDDP). The experimental steps of the MTT assay have been previously described [17,19]. Each experiment in the MTT assay was performed in triplicate to ensure repeatability and validity of the results.

### 2.4. Cell Proliferation Assessment—BrdU Test

A ready-for-use ELISA BrdU Kit from Roche Diagnostics (Mannheim, Germany) was used to assess cell proliferation. The test was carried out in accordance with the manufacturer’s instructions provided. Initially, appropriate cell densities (as above) of melanoma and normal cells were plated on microtiter plates (NEST). After 24 h, the cells were exposed to increasing concentrations of DAP (2–200 µM). The next steps of the BrdU assay have been previously described [19,20].

### 2.5. Cell Cytotoxicity—LDH Test

The cytotoxicity of DAP was assessed using the LDH assay (Cytotoxicity Detection KitPLUS LDH, Roche Diagnostics, Mannheim, Germany) based on the measurement of lactate dehydrogenase activity released into the medium from damaged cells. Specified densities (as above) of all the studied melanoma and normal human cell lines were plated on microtiter plates (NEST). After 24 h, the medium was replaced with fresh medium supplemented with DAP (at concentrations of 2–200 µM). The LDH test was carried out after 72 h incubation with DAP according to the manufacturer’s instructions. A more detailed description of the steps of the LDH test has been previously described [17,19].

### 2.6. Isobolographic Analysis

To thoroughly characterize the pharmacodynamic interactions between DAP and the five cytostatics in the malignant melanoma cell lines (FM55P, A375, FM55M2 and SK-MEL28), we performed an isobolographic analysis of the MTT assay. On the basis of the results of the MTT test, the percentage of inhibition of cell viability in relation to the specific concentrations of the tested substances (DAP, MTX, DOCX, VEM, EPR and CDDP) was determined. Based on a computer-assisted log-probit method [21], dose-response effect graphs were drawn, and the log-probit concentration–response lines enabled the calculation of the IC_50_ values for every tested drug. The collaborative effect of two concentration–response lines was tested for the DAP + MTX, DAP + DOCX, DAP + VEM, DAP + EPR, DAP + CDDP combinations, as previously described [16,22,23,24]. All required calculations (including the IC_50_ values, SEM, N, test of parallelism between the studied drugs) were computed automatically in an MS Excel spreadsheet with formulas originally derived from the log-probit method [21], but these were modified and adapted to the in vitro conditions as described in our earlier studies [16,20,22,23,24]. If two concentration-response curves are parallel, the additivity on the isobologram is illustrated as a diagonal line connecting the IC_50_ values for the tested compounds when used alone. However, if the concentration-response curves are not parallel, the additivity on the isobologram is illustrated as an area bounded by two (upper and lower) lines of additivity [22,24]. From the experimentally denoted IC_50_ values (based on MTT assay) for the drugs administered alone, median additive inhibitory concentrations for the mixtures (IC_50 add_) (at the fixed ratio of 1:1) of DAP + MTX, DAP + DOCX, DAP + VEM, DAP + EPR, DAP + CDDP were calculated, as described previously [25,26]. More information on isobolographic analysis can be found in the literature [19,26,27,28]. At the final stage of the isobolographic analysis, polygonograms are presented to illustrate and summarize the types of interactions between daphnetin and the five cytostatics in human melanoma cells. On each graph, drugs from combinations are connected by a line. The color of the line symbolizes the interaction: black—additive, green—synergistic, and red—antagonistic. The polygonogram is widely applied in isobolographic studies [29].

### 2.7. Statistical Analysis

GraphPad Prism (version 8.0, San Diego, CA, USA) was used for statistical analysis of the data (from MTT, BrdU and LDH tests). The analyses were performed with a one-way ANOVA test followed by Tukey’s post-hoc test. Data were presented as means ± standard errors (SEMs). The median inhibitory concentration (IC_50_ and the IC_50 mix_) values for DAP and the five tested cytostatics (DAP, MTX, DOCX, VEM, EPR and CDDP) were calculated automatically with a computer-assisted log-probit method using an MS Excel spreadsheet, as described earlier [16,21,22,23,24]. The unpaired Student’s *t*-test with Welch’s correction was used to statistically compare the experimentally derived IC_50 mix_ values (for the two-drug mixtures: DAP + MTX, DAP + DOCX, DAP + VEM, DAP + EPR, DAP + CDDP) with their respective, theoretically calculated and presumed to be additive IC_50 add_ values, as recommended elsewhere [25,30]. All the isobolograms were drawn in the MS Excel spreadsheet.

## 3. Results

### 3.1. Cell Viability Assessment—MTT Assay Results

All the tested compounds—daphnetin (DAP), mitoxantrone (MTX), docetaxel (DOCX), vemurafenib (VEM), epirubicin (EPR) and cisplatin (CDDP)—inhibited melanoma cell viability in a concentration-dependent manner. The main compound tested was the simple coumarin daphnetin, and its effect on viability in four human melanoma cell lines as well as in keratinocytes and melanocytes was assessed after 72 h incubation with DAP at concentrations of 2–200 µM (Figure 1). The melanoma cell lines FM55M2 (Figure 1c) and FM55P (Figure 1a) are very sensitive to DAP, because a significant reduction in their viability was already observed at the low concentrations of 2 and10 µM, respectively. In the case of the SK-MEL28 line (Figure 1d), a significant inhibition of cell viability was observed at a concentration of 50 µM; for the A375 line (Figure 1b), this was seen at 100 µM. In the case of normal cell lines, only a slight inhibition of cell viability was observed. More specifically, in the case of HaCaT keratinocytes (Figure 1e), inhibition of viability was observed for DAP at a concentration of 100 µM; however, at the highest tested concentration of 200 µM, viability was reduced by approximately 25%. In HEMa-LP normal melanocytes (Figure 1f), statistically significant growth inhibition was observed at 20 µM; however, the reduction in viability, even at the highest concentration of DAP (200 µM), did not exceed 15% (Figure 1f).

### 3.2. Cell Proliferation Assessment—BrdU Assay Results

All the tested malignant melanoma cells and normal cells were treated with increasing concentrations of DAP, displaying a concentration-dependent decrease in DNA synthesis (Figure 2). Measurements were made by evaluating the binding of BrdU (5-bromo-2′-deoxyuridine) to cellular DNA in proliferating cells. The results show that the cell proliferation of HaCaT keratinocytes (Figure 2e) and HEMa-LP melanocytes (Figure 2f) was inhibited only to a very small extent at DAP concentrations above 100 µM and 200 µM, respectively. In malignant melanoma cells, statistically significant inhibition of proliferation was observed at 10 µM DAP for the FM55P line (Figure 2a) and at 40–50 µM for the other cell lines tested (Figure 2b–d). At the highest tested concentration of DAP (200 µM), a complete inhibition of proliferation was observed for the FM55P and A375 primary and FM55M2 metastatic cell lines. In only the SK-MEL28 line, DAP at a concentration of 200 µM inhibited proliferation by about 80%.

### 3.3. Cytotoxicity of Daphnetin Assessment—LDH Assay Results

The cell cytotoxicity of DAP was assessed by the LDH assay, which measures lactate dehydrogenase release into the medium (Figure 3). The release of LDH indicates damage to the cell membrane and cell death [31]. DAP was cytotoxic to all the tested human melanoma cell lines at a concentration of 60 µM (Figure 3a–d). In the range of tested DAP concentrations of 2–200 µM, DAP was not cytotoxic to keratinocytes (Figure 3e), while to melanocytes, it was cytotoxic only at concentrations above 150 µM (Figure 3f).

### 3.4. Interactions between Daphnetin and One Chemotheraputic: Mitoxantrone, Docetaxel, Vemurafenib, Epirubicin or Cisplatin

The inhibitory effects of the tested cytostatics—MTX, DOCX, VEM, EPR and CDDP—in combination with DAP on the viability of malignant melanoma cells were analyzed in the MTT assay. The main purpose of this work was to determine whether DAP could enhance the anti-cancer effects of the tested chemotherapeutic drugs. Concentration-response curves (CRC) were determined according to the log-probit method [21]. From the equations of the curves determined for the compounds used alone and for the combination of DAP with one of the tested drugs—MTX, DOCX, VEM, EPR or CDDP (see Appendix A)—it was possible to determine the median inhibitory concentrations (IC_50_ values ± SEM). In our previous experiments, we had determined the mean inhibitory concentration (IC_50_) values for MTX (0.04 to 1.7 µM) [19], CDDP (1.3 to 3.3 µM) [16,19] and DOCX (from 1.27 to 15.83 nM), which varied depending on the cell line [17]. The IC_50_ values for EPR and VEM, respectively, were also determined for the following cell lines: FM55P (0.29 ± 0.07 µM and 0.76 ± 0.26 µM), A375 (0.26 ± 0.05 µM and 6.07 ± 2.06 µM), FM55M2 (0.16 ± 0.03 µM and 0.62 ± 0.27 µM) and SK-MEL28 (0.39 ± 0.08 µM and 0.25 ± 0.13 µM) [32]. The IC_50_ of DAP was determined at 40.48 ± 10.90 µM in the FM55M2 cell line, 64.41 ± 9.02 µM in the FM55P cell line, 116.59 ± 18.35 µM in the SK-MEL28 cell line and 183.97 ± 18.82 µM in the A375 cell line.

Parallelism of the experimentally determined curves of the concentration-response effects for all the tested combinations—DAP + MTX, DAP + DOCX, DAP + VEM, DAP + EPR and DAP+CDDP (see Appendix A)—was assessed using the log-probit method, as described earlier [21]. For the combinations of DAP + VEM and DAP+DOCX, the concentration-response curves are not parallel, except for the FM55M2 cell line (Appendix A). For the combinations of DAP+EPR and DAP + CDDP, the concentration-response curves are parallel, except for the A375 cell line (Appendix A). For the DAP + MTX combination, the concentration-response curves are not parallel for both the FM55P and A375 cell lines (Appendix A) but parallel for the FM55M2 and SK-MEL28 cell lines (Appendix A). Whether the lines are non-parallel or parallel, this fact affects the appearance of the additivity lines on the isobologram. For all the tested combinations (DAP + MTX, DAP + DOCX, DAP + VEM, DAP + EPR and DAP + CDDP), dose-response plots were drawn using GraphPad Prism (version 8.0) based on the results of MTT tests for the combinations of active substances in a fixed dose ratio of 1:1 (see Appendix A).

The isobolographic analysis of the combination of DAP with VEM (at the fixed ratio of 1:1) showed a statistically significant antagonistic interaction in the A375 and FM55M2 cell lines (Figure 4c,e) and an additive interaction with a tendency toward antagonism in the FM55P and SK-MEL28 cell lines (Figure 4a,g). An antagonistic interaction was observed for the combination of DAP with EPR in the four studied malignant melanoma cell lines (Figure 4b,d,f,h).

In the isobolographic analysis, the combination of DAP with CDDP (Figure 5a,c,e,g) and DAP with DOCX (Figure 5b,d,f,h) (both at the fixed ratio of 1:1) showed an additive interaction in all the tested human malignant melanoma cell lines.

Synergy was observed for the combinations of DAP with MTX in the FM55M2 and SK-MEL28 metastatic melanoma cell lines (Figure 6c,d). The combination of DAP with MTX (at the fixed ratio of 1:1) for primary melanoma cell lines (FM55P and A375) showed an additive interaction with a tendency toward synergy and additive interaction, respectively (Figure 6a,b).

A polygonogram (Figure 7) was drawn to graphically summarize the interactions for the two-drug mixture of daphnetin (DAP) and each of the five tested cytostatics (MTX, DOCX, VEM, EPR or CDDP). Illustration of the interactions in form of polygonograms makes it easy to evaluate the obtained interactions between drugs, which allows for faster determination of the most advantageous combinations. Our experiments showed that the most advantageous combination for the metastatic melanoma cell lines was the combination of DAP with MTX (Figure 7c,d).

## 4. Discussion

The results of our study confirmed that daphnetin dose-dependently inhibits the proliferation and viability of all the tested human malignant melanoma cell lines (FM55P, A375, FM55M2 and SK-MEL28). Researchers have confirmed the inhibitory effect of daphnetin on the viability of the murine metastatic osteosarcoma LM8 cell line. Daphnetin at doses of up to 30 µM had no effect on cell viability; however, the drug at a dose of 100 µM exerted an 80% inhibition of cell viability [33]. In laboratory experiments, it was noted that daphnetin inhibits protein kinases specific to tyrosine, threonine, protein kinase C and cAMP-dependent kinase. These protein kinases play an important role in cell metabolism, proliferation and differentiation. For instance, tyrosine kinase inhibition by daphnetin was competitive with ATP against the EGF receptor [34]. In the human hepatocellular carcinoma cell line HepG2, daphnetin did not inhibit EGF-induced tyrosine phosphorylation of the EGF receptor [34]. In addition, the anticancer potential of daphnetin is associated with its immunomodulatory capacity. This coumarin improves the direct cytotoxicity of NK cells against tumor cells in the presence of IL-12. In vitro experiments on purified human primary NK cells confirmed that daphnetin, in the presence of IL-12, directly activated the production of IFN-γ in NK cells through PI3K–Akt–mTOR signaling [35]. Kumar et al. in both in vitro and in vivo experiments confirmed a potential of daphnetin in inhibiting the angiogenesis process. Human umbilical vein endothelial cells (HUVEC) were treated with daphnetin (at concentrations of 9.375–900 mM), and inhibition of proliferation was observed for daphnetin at concentrations above 300 mM [36]. The anti-angiogenic effect of daphnetin was confirmed in the chicken chorioallantoic membrane test and the rat aortic ring (RAR) test induced by the vascular endothelial growth factor (VEGF) [25]. Daphnetin at concentrations of 5–40 µg/mL (≈28–224 µM) inhibited viability in three ovarian cancer cell lines [37]. Additionally, in the xenograft mouse model of the A2780 cell line (BALB/c mice), it was confirmed that daphnetin promoted the apoptosis of tumor cells without causing side effects in the tested mice [37]. In in vitro studies on two hepatocellular carcinoma cell lines (Huh7 and SK-HEP-1), daphnetin inhibited cell viability and promoted apoptosis. The obtained IC_50_ doses of daphnetin against the tested hepatocellular carcinoma lines Huh7 and SK-HEP-1 were 69.41 and 81.96 μM, respectively [38]. The IC_50_ values of daphnetin in this study ranged from 40.48 ± 10.90 µM to 183.97 ± 18.82 µM and are comparable to those obtained against hepatocellular carcinoma. A summary of the mechanisms of action of daphnetin in the in vitro model is presented in Table 1.

In vivo experiments on rats confirmed the anticancer effect of daphnetin in inhibiting liver cancer through antioxidant and anti-inflammatory effects. More specifically, Swiss Wistar rats were given diethylnitrosamine and phenobarbital to induce and progress hepatocellular carcinoma; then, the animals were given various doses of daphnetin to determine the effect of the compound on neoplastic lesions. The assessment of biochemical parameters showed a decrease in the levels of, among others, alkaline phosphatase, aspartate aminotransferase and total bilirubin, and an increase in the level of glutathione, catalase and superoxide dismutase. This proves that daphnetin alleviated the liver damage caused by diethylnitrosamine. In addition, daphnetin showed an anti-inflammatory effect that manifested as a decrease in the level of interleukins (IL-1β and IL-6), tumor necrosis factor-α (TNF-α), cyclooxygenase-2 and nuclear factor kappa B (NF-κB) [40]. The strong anti-inflammatory and anti-leukemic effect of daphnetin was demonstrated in albino Wistar rats which were given benzene to induce leukemia. In animals receiving daphnetin (at doses of 12.5 or 50 mg/kg body weight), the levels of pro-inflammatory cytokines, including TNF-α, NF-κB and interleukins (IL-1β, IL-2, IL-6), decreased. Daphnetin reduced the production of ROS (under the influence of benzene) and thus protected rats against the development of leukemia [41]. Moreover, daphnetin has the potential to prevent osteoporosis as demonstrated in rats given intramuscular dexamethasone to induce glucocorticoid-induced osteoporosis. Treatment with daphnetin increased bone mineralization and restored normal levels of bone turnover markers [42]. Daphnetin showed antitumor activity against 7,12-dimethylbenz(a)anthracene (DMBA)-induced breast cancer in female Sprague-Dawley rats. Daphnetin had strong antioxidant properties related to the protection against lipid peroxidation and the increase in antioxidant markers, both in serum and in rat organs—kidneys, liver and breast tissue. The anticancer mechanism of daphnetin resulted from the inhibition of p-AKT and the decrease in NF-κB expression [43]. In the same model of breast cancer in rats (DMBA-induced), it was shown that daphnetin (at doses of 20, 40 and 80 mg/kg body weight) prevented tumor growth due to the fact that the tumor became less sensitive to estrogens [42]. Significant decreases in the expression of p-MAPK1/2, p-JNK1/2, p-Akt, EGFR, IGFR, ERα, PR and aromatase were also observed [44].

In the case of studies on the effectiveness of daphnetin against melanoma, the experiments conducted on murine B16F10 cells confirmed its low cytotoxicity at doses of 1–50 µM [45]. Our experiments conducted on four different human malignant melanoma cell lines revealed that the cytotoxicity of daphnetin (assessed by the LDH test) was observed at concentrations above 60 µM. However, it was found that daphnetin diminished the synthesis and secretion of melanin by inhibiting the activity of cellular tyrosinase, which was associated with the modulation of the PKA/CREB and ERK/MSK1/CREB pathways, leading to a decrease in the expression of tyrosinase, Trp1 and Trp2 [45].

In another experiment, daphnetin at a concentration of 160 µM evoked cytotoxicity in the mouse melanoma (B16), mouse colon cancer (C26) and mouse breast adenocarcinoma (MXT) cell lines [46]. The IC_50_ value of daphnetin in B16 murine melanoma cells was 54 ± 2.8 µM [46], which is comparable to the IC_50_ values obtained in our experiments. For the mouse breast adenocarcinoma cells, the IC_50_ value for daphnetin was 74 ± 6.4 µM and for the mouse colon cancer cells, the IC_50_ value amounted to 108 ± 7.3 µM [46]. The inhibitory effect of daphnetin on melanoma tumor growth was confirmed in in vivo studies of C57BL/6 mice (B16 cell tumor) administered daphnetin at doses of 10, 20, and 40 mg/kg b.w. [46]. The higher the dose of this coumarin and the longer the time of its administration, the greater the inhibition of tumor growth observed compared with the control [46].

Importantly, daphnetin is safe to normal cells, which was confirmed in studies on normal ovarian cells (IOSE80), where concentrations of daphnetin (5 to 40 µg/mL ≈ 28–224 µM) were tested [37]. Our experiments confirm that daphnetin is not cytotoxic to keratinocytes and melanocytes up to a concentration of 150 µM. Above this concentration, a slight cytotoxicity of daphnetin was observed. Other in vitro experiments confirmed that low concentrations of daphnetin (up to 100 µM) did not affect the viability of normal cells such as normal liver cells (WRL68 line), primary hepatocytes (PH), human normal intestinal epithelial cells (NCM460) and gastric mucosal cells (GES-1) [38]. In the case of osteoblasts, daphnetin showed no cytotoxicity, but it enhanced cell proliferation, differentiation and mineralization by activating the Wnt/GSK-3β/β-catenin signaling pathway [42].

The maximum concentrations (Cmax) of the tested cytostatics in the patients’ serum are as follows: MTX (0.715 µM), DOCX (5.47 µM), VEM (127 µM), EPR (16.6 µM) and CDDP (14.4 µM) [47]. In the case of docetaxel, vemurafenib, epirubicin and cisplatin, the IC_50_ values we obtained are much lower than the Cmax. Only for mitoxantrone was the Cmax value exceeded for the SK-MEL28 cell line, for which the IC_50_ was 1.74 ± 0.51 µM. Effective cancer therapy involves the use of a number of therapeutic methods, such as surgical methods, radiotherapy, chemotherapy or immunotherapy, depending on the localization and stage of the cancer. For most of the drugs used, numerous side effects are observed, and drug resistance appears after some time of use. Multidrug resistance is also becoming more and more common, which further complicates chemotherapy [48,49,50]. Therefore, drug–drug combinations or combinations of drugs with naturally occurring compounds with anticancer potential are sought, which could improve the response to therapy. Drugs with different mechanisms of action are most often combined because they provide the opportunity for a synergistic interaction and, thus, intensification of their anti-cancer effect. It would also be desirable to simultaneously weaken the side effects, which is why combinations with plant-derived compounds are being tested. To date, scientists have evaluated the combination of the highly nephrotoxic cisplatin and daphnetin in in vitro and in vivo experiments. In human renal proximal tubular cells (HK-2), pretreatment with daphnetin (at doses of 2.5–10 μg/mL ≈ 14–56 µM) prior to the administration of cisplatin attenuated its cytotoxicity against normal cells. Daphnetin increased the expression of antioxidant enzymes and SIRT1, SIRT6 andNrf2 proteins, limiting the production of ROS induced by the cisplatin administered later [48]. In experiments on mice, the effectiveness of daphnetin in protecting the kidneys from cisplatin-induced damage was confirmed. The protective effect of daphnetin was associated with the regulation of antioxidant enzymes, an increase in the expression of SIRT1, SIRT6, Nrf2, HO-1 and NQO1, and a decrease in the expression of NOX4. Daphnetin restored normal creatinine levels, reversed weight loss in mice and attenuated the renal tubular necrosis caused by acute kidney injury as a result of cisplatin treatment [48]. Co-administration of cisplatin and daphnetin has also been evaluated in non-small-cell lung cancer (A549 and H1299) and ovarian cancer (A2780, OVCAR-8 and SKOV3) cell lines [48]. The conducted experiments revealed that the administration of daphnetin did not weaken the anticancer effect of cisplatin to any extent, which may suggest an additive nature of the interaction between these drugs [48]. Our experiments and isobolographic analysis showed additive interactions between daphnetin and cisplatin in all the tested melanoma cell lines (FM55P, A375, FM55M2, SK-MEL28). Importantly, daphnetin enhanced the anticancer effect of cisplatin, which was confirmed in studies on C57BL/6 mice injected with B16 murine melanoma cells [48]. An additive interaction was also observed for the combinations of fraxetin with cisplatin [20] and osthole with cisplatin [17] against melanoma cells. Due to the additive nature of the interaction of daphnetin with cisplatin and the protective effect on normal cells, which allows the reduction of side effects of cytostatics, these combinations are worthy of further research. The results of our experiments also showed an additive interaction in melanoma cells for the combination of daphnetin with docetaxel.

Deng et al. tested a combination of daphnetin with doxorubicin in in vitro tests on human esophageal squamous cell carcinoma cells (YM1 line). Daphnetin arrested the cell cycle in S phase, with an IC_50_ value of 253 μM [49]. In a mouse xenograft study of esophageal cancer, the combination of daphnetin and doxorubicin had lower systemic toxicity compared with doxorubicin alone. In addition, the drugs given in combination increased the body weight of the mice and, most importantly, reduced tumor size more than the chemotherapeutic agent used alone [49]. Our experiments with the combination of daphnetin and epirubicin (a derivative of doxorubicin) showed antagonistic interactions in the four malignant melanoma cell lines. The observed difference in interactions may result from the response of the tumor itself to a given drug or combination, as well as from differences in the structure and action of epirubicin and doxorubicin. Of note, antagonism is an undesirable type of interaction due to the abolition of the therapeutic effect of the co-administered drugs. Our experiments have shown that the combination of daphnetin with the BRAF inhibitor vemurafenib showed additivity with a tendency toward antagonism or an antagonistic interaction in human melanoma cells.

From the pharmacological point of view, the most desirable interaction is synergy, which proves the mutual intensification of the therapeutic effect of drugs in combination. Daphnetin in combination with mitoxantrone showed synergy in both metastatic human melanoma cell lines (FM55M2 and SK-MEL28), while in primary melanoma cell lines, the combination of daphnetin with mitoxantrone showed an additive interaction (A375) or additivity with a tendency toward synergy (FM55P).

## 5. Conclusions

Daphnetin, at lower concentrations of up to 100–150 µM, is safe for keratinocytes and melanocytes and does not limit their proliferation or viability. Above these concentrations, it causes only slight inhibition of the growth of normal cell lines. However, daphnetin exhibits anticancer activity by inhibiting viability and proliferation of the tested human malignant melanoma cell lines. The IC_50_ values of daphnetin range from 40.48 ± 10.90 µM to 183.97 ± 18.82 µM, depending on the cell line. Isobolographic analysis showed that the interaction for the combination of daphnetin with vemurafenib or epirubicin was antagonistic. An additive interaction was observed for the combinations of daphnetin with cisplatin and docetaxel. The most desirable synergistic interaction was observed for the combination of daphnetin with mitoxantrone in metastatic human melanoma lines (FM55M2 and SK-MEL28), while in primary melanoma lines, this combination showed an additivity with a tendency toward synergy. The interactions found in this study may have a clinical significance, but further research is needed to understand the mechanisms of action of the combinations, and additional pre-clinical studies on animals should shed more light on malignant melanoma treatment with daphnetin.

## Figures and Tables

**Figure 1 cells-12-01593-f001:**
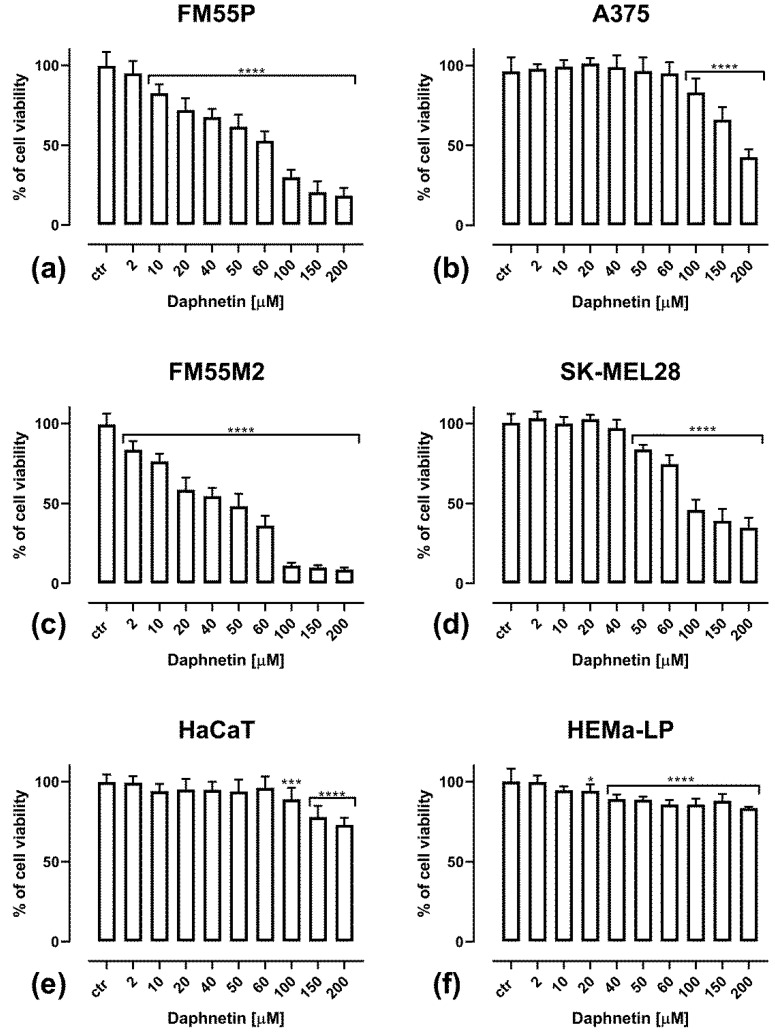
Impact of DAP on cell viability in the MTT test in FM55P (**a**), A375 (**b**), FM55M2 (**c**), SK-MEL28 (**d**), model of human keratinocyte HaCaT (**e**) and normal human melanocyte HEMa-LP (**f**) cell lines. Columns represent mean ± SEM (**** *p* < 0.0001, *** *p* < 0.001 and * *p* < 0.05 vs. the control (ctr) group).

**Figure 2 cells-12-01593-f002:**
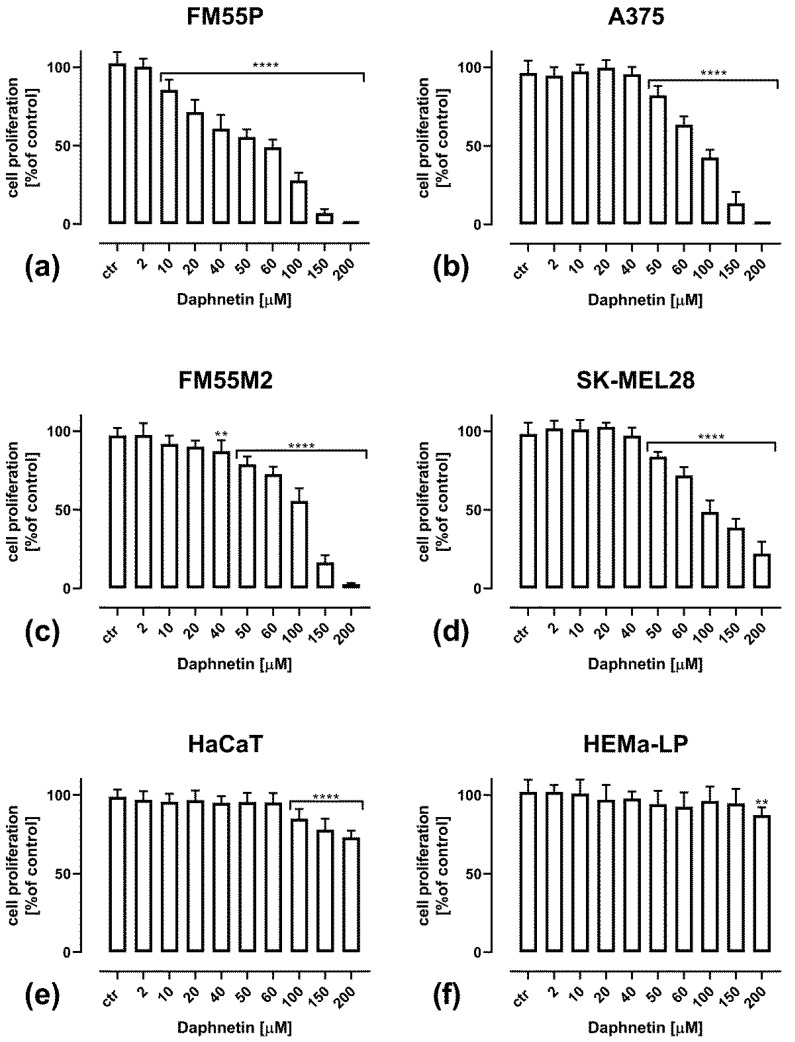
Impact of DAP on cell proliferation in the BrdU assay in FM55P (**a**), A375 (**b**), FM55M2 (**c**), SK-MEL28 (**d**), HaCaT human keratinocyte (**e**) and normal human melanocyte HEMa-LP (**f**) cell lines. The columns represent the mean for each concentration ± SEM. (**** *p* < 0.0001, and ** *p* < 0.01 vs. the control (ctr) group).

**Figure 3 cells-12-01593-f003:**
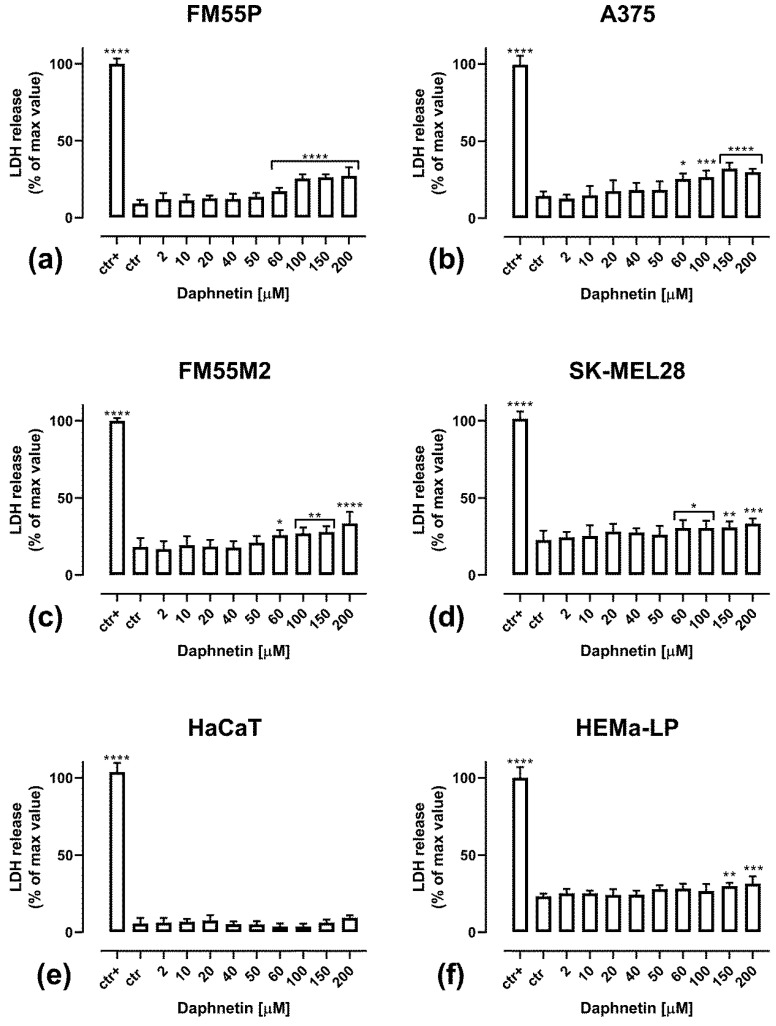
Impact of DAP on cytotoxicity in the LDH assay in FM55P (**a**), A375 (**b**), FM55M2 (**c**), SK-MEL28 (**d**), human keratinocyte HaCaT (**e**) and normal human melanocyte HEMa-LP (**f**) cell lines. ctr, cells in control medium; ctr+, cells treated with lysis buffer. The columns represent the mean for each concentration ± SEM. (**** *p* < 0.0001, *** *p* < 0.001, ** *p* < 0.01 and * *p* < 0.05 vs. the control (ctr) group).

**Figure 4 cells-12-01593-f004:**
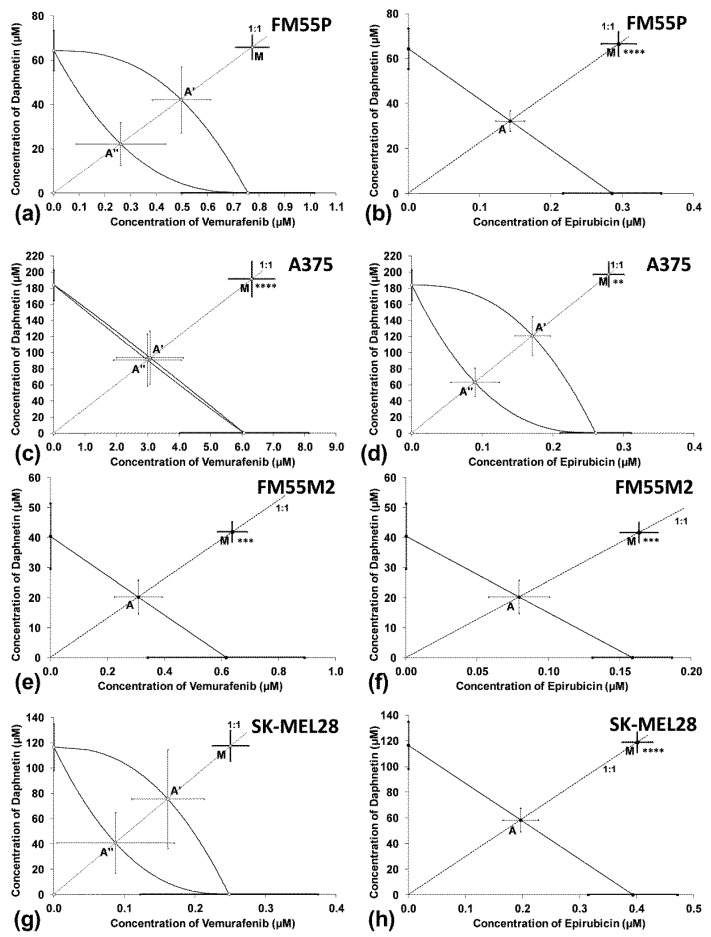
Interactions between DAP and VEM (**a**,**c**,**e**,**g**) and DAP and EPR (**b**,**d**,**f**,**h**) with respect to their anti-proliferative effects on FM55P (**a**,**b**), A375 (**c**,**d**), FM55M2 (**e**,**f**) and SK-MEL28 (**g**,**h**) malignant melanoma cells. The IC_50_ (±SEM) values for DAP, VEM and EPR are placed in the Cartesian system of coordination. Points A, A’, A”—theoretically calculated IC_50add_ values (±SEM). Point M—experimentally derived IC_50mix_ value (±SEM). **** *p* < 0.0001, *** *p* < 0.001 and ** *p* < 0.01 (Student’s *t*-test with Welch’s correction).

**Figure 5 cells-12-01593-f005:**
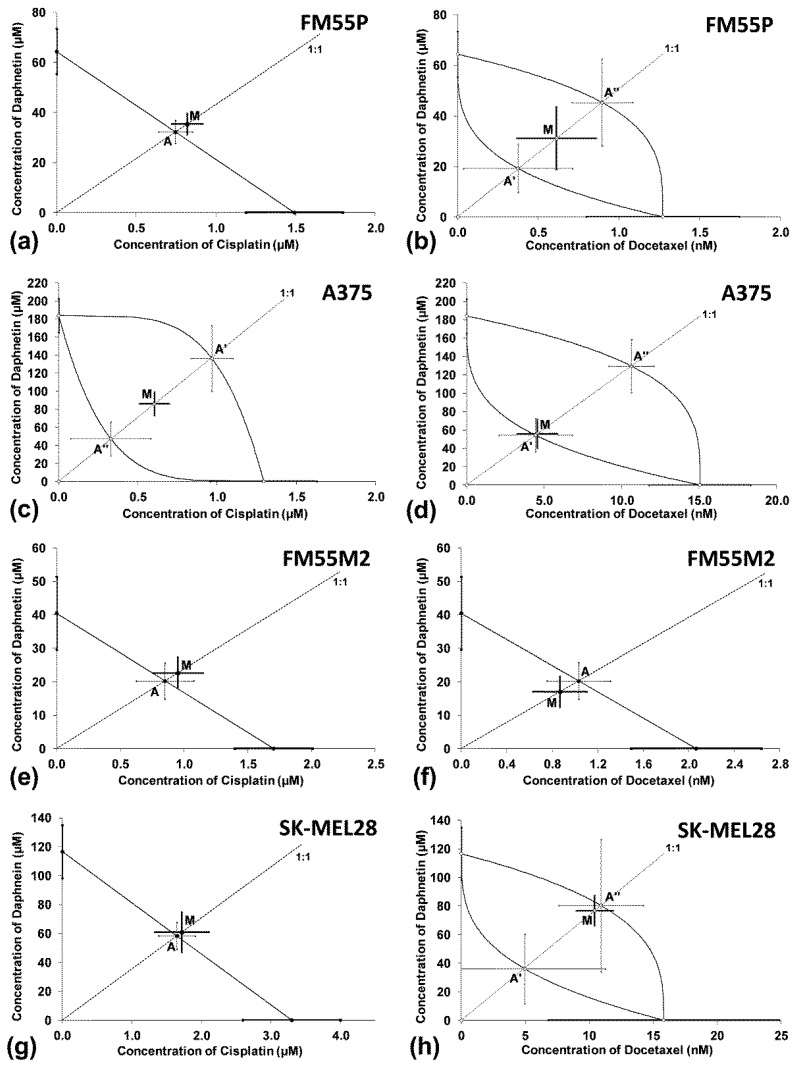
Interactions between DAP and CDDP (**a**,**c**,**e**,**g**) and DAP and DOCX (**b**,**d**,**f**,**h**) with respect to their anti-proliferative effects on FM55P (**a**,**b**), A375 (**c**,**d**), FM55M2 (**e**,**f**), and SK-MEL28 (**g**,**h**) malignant melanoma cells. The IC_50_ (±SEM) values for DAP, CDDP and DOCX are placed in the Cartesian system of coordination. Points A, A’, A”—theoretically calculated IC_50add_ values (±SEM). Point M—experimentally derived IC_50mix_ value (±SEM).

**Figure 6 cells-12-01593-f006:**
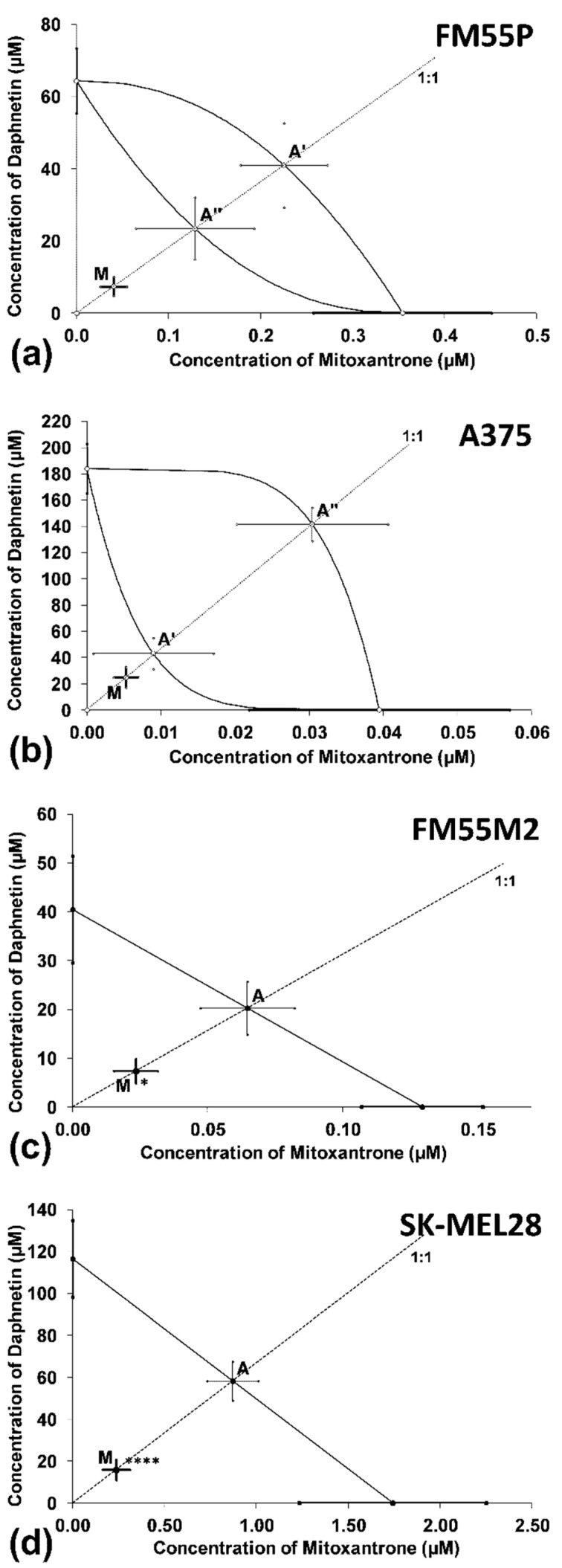
Interactions between DAP and MTX (**a**–**d**) with respect to their anti-proliferative effects on FM55P (**a**), A375 (**b**) FM55M2 (**c**) and SK-MEL28 (**d**) malignant melanoma cells. The IC_50_ (±SEM) values for DAP and MTX are placed in the Cartesian system of coordination. Points A, A’, A”—theoretically calculated IC_50add_ values (±SEM). Point M—experimentally derived IC_50mix_ value (±SEM). **** *p* < 0.0001 and * *p* < 0.05 (Student’s *t*-test with Welch’s correction).

**Figure 7 cells-12-01593-f007:**
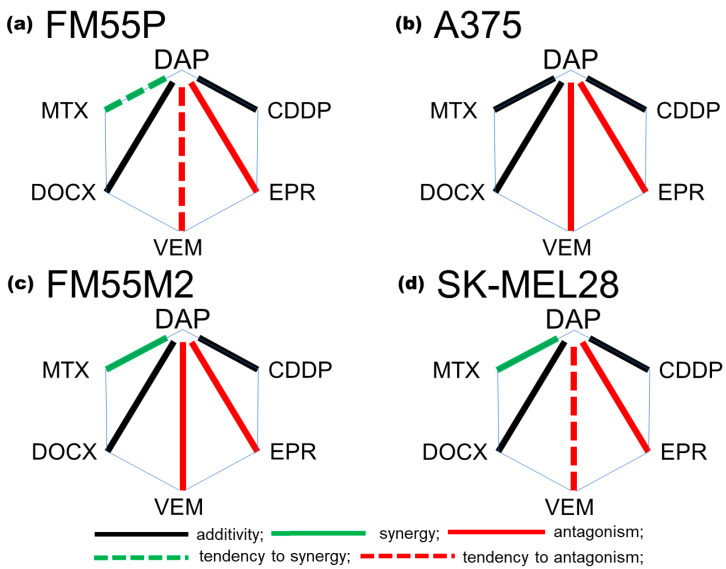
Polygonograms illustrating the interactions for two-drug mixtures of DAP with five different cytostatics (MTX, DOCX, VEM, EPR or CDDP) in in vitro tests for four malignant melanoma cell lines: FM55P (**a**), A375 (**b**) FM55M2 (**c**) and SK-MEL28 (**d**). Black lines illustrate the additive interactions, whereas the green lines indicate synergistic and red lines antagonistic interactions. The dashed lines indicate a tendency toward a given interaction.

**Table 1 cells-12-01593-t001:** Summary of the mechanisms of action of daphnetin in in vitro studies.

Type of Cancer	Cell Line	Dose of Daphnetin	Mechanism of Action	References
Murine metastatic osteosarcoma	LM8	≥100 μM	reduced levels of the proteins that regulate actin polymerization, RhoA and Cdc42 (which are responsible for the production of stress fibers, filopodia and lamellipodia), thereby inhibiting cell invasion and migration	[33]
Ovarian cancer	A2780, SKOV3, OVCAR8	5–40 µg/mL (≈28–224 µM)	increased amounts of reactive oxygen species (ROS); increased expression of pro-apoptotic proteins (Bax, PARP and caspase-3) and reduced expression of Bcl2 (anti-apoptotic protein);	[37]
			induced cell autophagy through abnormalities in the functioning of the AMPK/Akt/mTOR pathway	
Hepatocellular carcinoma	Huh7, SK-HEP-1	≥100 μM	cell cycle arrest in the G1 phase	[38]
			inhibition of Wnt/β-catenin signaling	
Human renal adenocarcinoma	A-498	≥500 μM	cell cycle arrest in the S phase (at low dose, 48 h and 72 h incubation) or G1 phase (250 and 500 µM, 96 h);	[39]
			activation of mitogen-activated protein kinase p38 MAPK and inhibition of ERK1/ERK2	

## Data Availability

Data are contained within the article.

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
