# Peer review of "Daphnetin, a Coumarin with Anticancer Potential against Human Melanoma: In Vitro Study of Its Effective Combination with Selected Cytostatic Drugs"

_cells, 2023, doi:10.3390/cells12121593_

Round 1

Reviewer 1 Report

Authors should provide more clinical information and potential application of therapies/treatments in this manuscript.

Author Response

Comments:

Authors should provide more clinical information and potential application of therapies/treatments in this manuscript.

Reply:

More clinical information has been added, among others, maximum serum concentrations for the tested cytostatics. Information on the potential application of the acquired knowledge in the clinical situation has been added. Further research is required, primarily based on preclinical studies on animals.

Reviewer 2 Report

1.   Title “Hu-Man” should be human.

2.   What are normal serum concentrations that are achieved in patients with each of compounds, or their active metabolites?  Are they similar to the ranges used in the current study?  If not, discuss in limitations.

3.   I didn’t see mention of the mutation profile of the four cell types, which are all V600E, and justifies the use of vemurafinib. It should be made clear that the A375 cells are the original, not the CRISPR-modified versions that are also sometimes sold under the same name “A375” from ATCC

4.   Results- were the combination drug treatments performed in the current study or the previous study that they referenced?  If the dose response was performed in this study, then all data points should be shown, along with Stand Dev.

5.   In the text, the authors refer to other studies, some of which give the doses in w/v and some which give molarity.  All should be converted to molarities. 

6.   LDH assays are given as cytotoxicity, whereas MTT assays are given as survival.  However, this both assays can be used to determine survival or cytotoxicity.   So, additivity vs. synergy should be the same for both assays. 

7.   HaCaT are not normal human keratinocytes.  They contain heterozygous p53 gain of function mutations, and therefore may be more resistant to toxic agents (especially chemotherapy/genotoxic agents)

8.   Much of this methodology is explained very briefly by referring to the paper by Litchfield and Wilcoxon, 1948.  This paper in turn refers to testing Bliss and other models.  More explanation is needed to describe how these methods were adapted to the current study, and which of these are novel or new.  At least this should be updated into the 21st century use of computers rather than using graph paper and a straight edge as in Litchfield et al. Otherwise, what is the purpose of plugging in combinations of drugs into a formula derived in the Litchfield paper?  Is it the use of the drugs themselves that are novel, or some innovative adaptation of the Litchfield paper?  Does this tell us anything about the potential mechanisms of action of the different compounds?  Are all these compounds even being used for melanoma in the clinic, alone or in combination?  If so, these need references and serum concentrations of compounds and metabolites so one can get a rough idea of the real-world feasibility of the study.

9.   Related to the above concern, is there any hypothesis concerning how different DNA damage- and adduct-inducing compounds might behave differently in this assay? While there is a brief discussion of B16 mouse melanoma cells and pigmentation, there is no insight into the mechanism of cytotoxicity/proliferation/survival and its synergy, additivity, or antagonism.  It certainly is important to know that certain combinations might be avoided, but beyond this it needs to relate to the use of these drugs in the clinic currently (beyond vemurafinib, mentioned above).  What about vemurafinib plus dabrafenib which has been used in combination for the last 10 years- how does this work in the model?  Would cyp inhibitors change the observed interactions?

10.                 The values and SD of supplementary graphs are difficult to read.  The significance and discussion of parallelism needs to be more developed in the manuscript.

Minor corrections

Author Response

  1. Title “Hu-Man” should be human.

The hyphenation error in the title has been corrected, as suggested.

  1. What are normal serum concentrations that are achieved in patients with each of compounds, or their active metabolites?  Are they similar to the ranges used in the current study?  If not, discuss in limitations.

Maximum concentrations (Cmax) of the tested cytostatics in the patients' serum have been added and compared with the IC50 values of the studied cytostatics, as suggested.

  1. I didn’t see mention of the mutation profile of the four cell types, which are all V600E, and justifies the use of vemurafinib. It should be made clear that the A375 cells are the original, not the CRISPR-modified versions that are also sometimes sold under the same name “A375” from ATCC

All the necessary information has been added in the Materials and Methods section, as suggested.

  1. Results- were the combination drug treatments performed in the current study or the previous study that they referenced?  If the dose response was performed in this study, then all data points should be shown, along with Stand Dev.

For all the tested two-drug combinations, dose-response effect plots were added to the Supplementary materials as Figures S4, S5 and S6, as suggested.

  1. In the text, the authors refer to other studies, some of which give the doses in w/v and some which give molarity.  All should be converted to molarities. 

Doses of the drugs presented in w/v were converted to molarity, as suggested.

  1. LDH assays are given as cytotoxicity, whereas MTT assays are given as survival.  However, this both assays can be used to determine survival or cytotoxicity.   So, additivity vs. synergy should be the same for both assays. 

Indeed, the LDH and MTT tests are similar in some aspects, but we assessed the effects of mixtures of the investigated compounds using the MTT test only.

  1. HaCaT are not normal human keratinocytes.  They contain heterozygous p53 gain of function mutations, and therefore may be more resistant to toxic agents (especially chemotherapy/genotoxic agents).

Appropriate changes have been incorporated into the text, as suggested.

  1. Much of this methodology is explained very briefly by referring to the paper by Litchfield and Wilcoxon, 1948.  This paper in turn refers to testing Bliss and other models.  More explanation is needed to describe how these methods were adapted to the current study, and which of these are novel or new.  At least this should be updated into the 21stcentury use of computers rather than using graph paper and a straight edge as in Litchfield et al. Otherwise, what is the purpose of plugging in combinations of drugs into a formula derived in the Litchfield paper?  Is it the use of the drugs themselves that are novel, or some innovative adaptation of the Litchfield paper?  Does this tell us anything about the potential mechanisms of action of the different compounds?  Are all these compounds even being used for melanoma in the clinic, alone or in combination?  If so, these need references and serum concentrations of compounds and metabolites so one can get a rough idea of the real-world feasibility of the study.

The log-probit method has been originally described by Litchfield and Wilcoxon [1949]. However, all the necessary calculations allowing for the determination of the IC50 values along with their SEM values were performed automatically in the MS Excel spreadsheet. All required calculations (including, the IC50, SEM, N, test of parallelism between the studied drugs) were computed automatically in the MS Excel spreadsheet with formulas originally derived from the log-probit method [1949], but modified and adapted to the in vitro conditions, as described in our earlier studies. Detailed information about equations, illustrating on how to compute the required values have been published earlier [Luszczki JJ, Ratnaraj N, Patsalos PN, Czuczwar SJ. Isobolographic analysis of interactions between loreclezole and conventional antiepileptic drugs in the mouse maximal electroshock-induced seizure model. Naunyn Schmiedebergs Arch Pharmacol. 2006 May;373(2):169-81. doi: 10.1007/s00210-006-0055-4.; Luszczki JJ. Isobolographic analysis of interaction between drugs with nonparallel dose-response relationship curves: a practical application. Naunyn Schmiedebergs Arch Pharmacol. 2007 Apr;375(2):105-14. doi: 10.1007/s00210-007-0144-z.]

Maximum concentrations (Cmax) of the tested cytostatics in the patients' serum have been added and compared with the IC50 values, as suggested.

  1. Related to the above concern, is there any hypothesis concerning how different DNA damage- and adduct-inducing compounds might behave differently in this assay? While there is a brief discussion of B16 mouse melanoma cells and pigmentation, there is no insight into the mechanism of cytotoxicity/proliferation/survival and its synergy, additivity, or antagonism.  It certainly is important to know that certain combinations might be avoided, but beyond this it needs to relate to the use of these drugs in the clinic currently (beyond vemurafinib, mentioned above).  What about vemurafinib plus dabrafenib which has been used in combination for the last 10 years- how does this work in the model?  Would cyp inhibitors change the observed interactions?

This is an accurate observation. We are going to determine the effects exerted by three-drug combinations in further studies, including the combinations of daphnetin + vemurafinib + dabrafenib.

  1. The values and SD of supplementary graphs are difficult to read.  The significance and discussion of parallelism needs to be more developed in the manuscript.

We are sorry that the charts turned out to be illegible. In the electronic version, they have a high resolution and are very readable.

Information about parallelism of curves has been added as suggested.

Reviewer 3 Report

The authors analyzed the mode of action of daphnetin in different melanoma cell lines in combinatorial therapy with other agents. Their data show synergistic but also antagonistic therapeutic effects of daphnetin with the other therapeutics. Interestingly, an antagonistic adverse therapy effect can also be observed with vemurafenib and epirubicin.
However, the authors also emphasize the synergistic effect between daphnetin and cisplatin and docetaxel.

The manuscript is very well written and the data are very very well discussed. The isobolographic presentation is very convincing.

Author Response

Thank you for reviewing our manuscript.